# Monitoring Cultured Rat Hepatocytes Using RNA-Seq In Vitro

**DOI:** 10.3390/ijms24087534

**Published:** 2023-04-19

**Authors:** Yung-Te Hou, Chia-Chun Wu, Wen-Ting Wang, Wen-Tse Yang, Ying-Hsiu Liao, Chien-Yu Chen

**Affiliations:** Department of Biomechatronics Engineering, National Taiwan University, Taipei 106, Taiwan

**Keywords:** RNA-Seq, hepatocyte culture, qPCR

## Abstract

Compared to other techniques, RNA sequencing (RNA-Seq) has the advantage of having details of the expression abundance of all transcripts in a single run. In this study, we used RNA-Seq to monitor the maturity and dynamic characteristics of in vitro hepatocyte cultures. Hepatocytes, including mature hepatocytes and small hepatocytes, were analyzed in vitro using RNA-Seq and quantitative polymerase chain reaction (qPCR). The results demonstrated that the gene expression profiles measured by RNA-Seq showed a similar trend to the expression profiles measured by qPCR, and can be used to infer the success of in vitro hepatocyte cultures. The results of the differential analysis, which compared mature hepatocytes against small hepatocytes, revealed 836 downregulated and 137 upregulated genes. In addition, the success of the hepatocyte cultures could be explained by the gene list screened from the adopted gene enrichment test. In summary, we demonstrated that RNA-Seq could become an effective method for monitoring the whole transcriptome of hepatocyte cultures and provide a more comprehensive list of factors related to the differentiation of small hepatocytes into mature hepatocytes. This monitoring system not only shows high potential in medical applications but may also be a novel method for the clinical diagnosis of liver-related diseases.

## 1. Introduction

Small hepatocytes, which are a subpopulation of hepatocytes and are considered a type of liver progenitor cell, have the potential to proliferate, maintain hepatic functions, and differentiate into mature hepatocytes [1]. Liver progenitor cells exist in fetal and adult livers and are activated for liver regeneration when severe liver injures occur (e.g., partial hepatectomy) or when mature hepatocytes lose their growth capabilities [2,3]. Small hepatocytes can express the phenotypic characteristics of fetal hepatoblasts, oval cells, and fully differentiated hepatocytes during the liver regeneration process [4]. Since liver development is a dynamic process, some analytic methods, such as clonogenicity and repopulation assays, have been performed to determine the potential of liver progenitor cells to differentiate into mature hepatocytes. Nevertheless, experiments involving the observation of cell lineage differentiation in vivo are practically difficult and are therefore limited to clinical applications [3]. Even today, both the differentiation and maturation processes of liver progenitor cells, as well as the molecular conditions of small hepatocytes during liver development, are not yet fully understood. This might be due to a lack of knowledge or limited biomarkers, which restricts trajectory tracking [3]. Furthermore, there are few reports that include quantification methods for liver differentiation in vitro [5]. Developing a faster and easier method with higher accuracy is very important for understanding the mechanisms of liver development and the generation process at a cellular level.

Recently, transcript expression analysis has become a commonly used method for phenotypic analysis; mRNA expression can be used to monitor the state of a cell in terms of its survival. Traditionally, microarrays have been used for this purpose; however, microarrays suffer from limitations such as lower accuracy and sensitivity when detecting certain transcripts, which occur due to problems with target-specific characteristics and the detection of dyeing substances. By contrast, RNA sequencing (RNA-Seq) has the potential to be a novel method for improving understandings of cell lineage relationships and heterogeneity in a given cell population [3,6,7]. RNA-Seq allows us to supplant coarse notions of marker-based cell types and uncover new cell types through the unbiased sampling of single cells [3,8]. As an emerging and flourishing technology, RNA-Seq can theoretically resolve some of the limitations that microarrays present, including the difficulties in designing microarrays for alternative RNA cleavage products and unknown gene targets. RNA-Seq has high-throughput- and non-target-specific characteristics, so it is frequently used to discover whole genes, transcripts, and genomic variations in various organisms. In particular, RNA-Seq has a higher sensitivity when analyzing differential gene expression levels [9]. As a substitution for microarrays, RNA-Seq has become the mainstream methodology for differential gene performance analysis, and has been widely tested for determining cell differentiation using transcript expression changes [10]. The analysis process is carried out by splicing the sequence reads into small segments via mapping and quantification, and there are various algorithms that can be used for this.

Differentiated hepatocytes are potential cellular sources for establishing liver models and testing clinical therapies. However, conventional methods, such as determining the expression levels of liver-specific lineage markers via quantitative polymerase chain reaction (qPCR), Western blot analysis, and immunocytochemical analysis, cannot provide a complete picture regarding the maturation and differentiation status of a cell or the degree of similarity between the whole liver and differentiated cell sources [5]. Previous researchers have already compared genome-wide gene expression analyses to individual liver cell models [11,12,13], but there are few reports comparing in vitro liver cell models to isolated liver tissue [13]. Therefore, we aimed to develop a method based on RNA-Seq analysis to assess the differentiation and maturation status of in vitro differentiated hepatocytes and overcome limitations related to the validation of this differentiation. If the intensities quantified by the RNA-seq method correlate with the observed developmental stages of in vitro hepatocyte cultures, RNA-Seq may be an effective method for the prediction of liver developmental stages in vivo.

In this study, small hepatocytes were analyzed in vitro and in silico. RNA-Seq analysis was used to investigate the maturation of small hepatocytes into mature hepatocytes from in vitro liver models in comparison to isolated hepatocytes. Moreover, qPCR, which was conducted using a selected set of important markers, was also run for comparison with the RNA-Seq analysis. We therefore examined the RNA expression levels in fresh isolated mature hepatocytes compared with the RNA expression levels in both fresh isolated and long-term-cultured small hepatocytes to analyze the differentiation of small hepatocytes both in vitro and in silico.

## 2. Results

### 2.1. Small Hepatocytes Have a Remarkable Ability to Grow and Proliferate In Vitro

The morphology of small hepatocytes and mature hepatocytes was assessed using phase-contrast microscopy (Figure 1 and Figure 2A) and fluorescence imaging (Figure 2B), respectively. The single or double nuclei of mature hepatocytes showed bright and regular contours with a spherical shape within three days of culturing (Figure 1A,B), indicating that the mature hepatocytes were undergoing growth and proliferation [14]. However, mature hepatocytes experienced progressive dedifferentiation after three days when cultured as a monolayer in vitro (Figure 1C,D), which eventually resulted in poor viability. This result is consistent with previous findings [15].

On the other hand, the small hepatocytes could proliferate, and exhibited multinuclear morphology after seven days (Figure 1E,F). In addition, the cells maintained their functional mesenchymal form even as the culture time increased, and they tended to form a colony pattern by day 13 (Figure 1G). The average size of the colonies increased with culture time (Figure 1G,H), and they were usually irregular in shape and contained 30–40 cells. The expansion of the colonies was clear, and there was little contamination by other nonparenchymal cells. Moreover, the cells typically stacked into three-dimensional spheroids during culturing, and retained their high cell viability after 35 days (Figure 2A). This spheroid formation has been recognized as important evidence that hepatocytes can maintain their functions [14]; here, they often retained the same pattern for more than three weeks (Figure 2). Fluorescent images confirmed that the colonies were alive after 35 days of culturing (Figure 2B).

### 2.2. Reproducibility of RNA-Seq

The pairwise correlation plots integrated in Figure 3 were derived by calculating the correlation coefficients between all pairs of RNA-seq samples (Table 1) based on 69,400 transcript expression intensities. M1 and M2 were two biological replicates directly retrieved from the rats (Rat a and Rat b) without culturing, S0-1 and S0-2 were two technical replicates taken on day zero, and S7-1 and S7-2 were two technical replicates taken on day seven. The correlation coefficients between the two pairs of technical replicates were obviously high (S0-1 and S0-2: 0.87; S7-1 and S7-2: 0.89). As shown in Table 1, Rat a and Rat b were two different Sprague Dawley rats and were used as biological replicates; whereas Rat A and Rat B were the other two different Sprague Dawley rats and were also used as biological replicates; S7-1, S7-2, S7-A, S14-A, and S21-A were from Rat A, whereas S7-B, S14-B, and S21-B were from Rat B. In Table 1, the twelve samples were sequenced in two batches, where the first batch sequenced the first six samples (M0, M1, S0-1, S0-2, S7-1, and S7-2) and the second batch sequenced the rest. The correlation coefficients of these two pairs of samples sequenced in different batches were also high (S7-1 and S7-A: 0.88; S7-2 and S7-A: 0.88), demonstrating the fidelity and reproducibility of the RNA-Seq analysis adopted in this study. Since S7-A was sequenced in the second batch, and S7-1 and S7-2 were sequenced in the first batch, we concluded that the batch effect was small.

### 2.3. RNA-Seq Analysis Was Consistent with qPCR Results

Albumin synthesis is a predominant hepatocyte-specific function studied in liver research. Follistatin (FO) expression can accelerate the proliferation of small hepatocytes [16], while tryptophan 2,3-dioxygenase (TO) is constitutively expressed in the liver [17]. Therefore, liver-specific functional genes for albumin (hepatic function), FO (small hepatocyte marker), and TO (mature hepatocyte marker) were targeted in this study. Hepatocyte gene expression levels for these hepatic-specific markers were analyzed by qPCR after 7, 15, and 21 days of culturing. In Figure 4, day 7 was used as the control to plot the trend. The quantities of other days were normalized to the average of day 7. Since Rat A and Rat B were plotted separately, there were no RNA-seq replicates.

According to the results of qPCR and RNA-Seq, in Rat A, the expression levels of albumin and TO were steady for 15 days, but decreased on day 21 (Figure 4A,C). FO expression slightly but continuously increased with culture time (Figure 4E). In contrast, in Rat B, the expression levels of albumin and TO increased within the first 15 days and decreased on day 21 (Figure 4B,D), and FO expression decreased on day 15 and increased on day 21 (Figure 4F). We therefore suggest that small hepatocytes in Rat B might have differentiated into mature hepatocytes because of the increased TO and decreased FO expression. The increased expression of albumin in Rat B also indicates that small hepatocytes exhibited more hepatic functions after long-term culturing.

Furthermore, the three target genes were used for comparing the qPCR and RNA-Seq analyses (Figure 4). All genes showed similar expression patterns, but there were some fold change differences between qPCR and RNA-Seq. For example, in Rat A, TO had decreased and albumin had increased after 15 days of culture according to the RNA-Seq analysis, but not the qPCR analysis (Figure 4A,C). Nevertheless, the qPCR and RNA-Seq results were generally consistent.

### 2.4. Differential Analysis Identified Thousands of Differentially Expressed Genes (DEGs)

For the RNA-Seq data analysis, the quality of the raw sequence files was checked by FastQC [18] and was found to meet the experimental criteria. For subsequent reference transcriptome mapping, detailed statistics of the mapping rate of each sample are shown in Table 2. The mapping (alignment) rate of each sample was 72.1–81.5%, which was sufficient for subsequent quantitative analysis. Preliminary statistics on the distribution of transcript expression levels after quantification are shown in Table 3. It was observed that the mature hepatocyte samples exhibited a smaller number of expressed transcripts. After conducting differentially expressed gene (DEG) analysis using DESeq2 [19] and comparing the mature hepatocytes (M) with the small hepatocytes (S0), 973 transcripts were obtained (adjusted *p* < 1 × 10^−5^; downregulated [log2-fold change < −1]: 836 (Appendix A); upregulated [log2-fold change > 1]: 137 (Appendix A)); the fold change is defined as the expression of M divided by that of S0.

Next, the 973 differentially expressed transcripts were sent for gene enrichment analysis. The Gene Ontology (GO) [20] enrichment analysis results, focusing on biological processes, are shown in Appendix A (downregulated) and Appendix A (upregulated). Appendix A shows that genes related to metabolic processes were expressed more in mature hepatocytes than in small hepatocytes. This is because liver progenitor cells are immature cells in nature, and therefore express lower levels of cytochrome P450 activity [21]. On the other hand, Appendix A demonstrates that small hepatocytes exhibited a higher expression of genes related to cell–cell adhesion (GO: 0007155; GO: 0098609) and actin cytoskeleton organization (GO: 0030036). On the other hand, Appendix A shows that mature hepatocytes exhibited comparatively higher expression levels of genes involved in the oxidation–reduction process (GO: 0055114), which is consistent with a previous study [22].

In Figure 5, we plotted 836 selected downregulated genes to observe their expression values in Rat A and Rat B. Rat B seemed to exhibit the expected trend, wherein the selected genes were downregulated as the small hepatocytes became more mature. In Figure 6, we plotted 137 selected upregulated genes to observe their expression values in Rat A and Rat B. Rat B also exhibited more expected gene behavior, wherein some of the selected genes were upregulated as the small hepatocytes became mature, even though the average did not change much.

## 3. Discussion

The development of liver cell-based therapies and further investigation into liver transplantation for patients with severe end-stage liver disease have been recognized as research priorities. Some alternative progenitor cell sources, such as induced pluripotent stem (iPS) cells, provide a potentially effective approach for both clinical transplantation and drug development applications [5]. However, mature hepatocytes differentiated from embryonic stem cells or progenitor cells have not yet matured to a stage where they could efficiently repopulate the liver in vivo [23]. Ascertaining a procedure for assessing liver progenitor cell maturation is thus very important for these clinical applications.

qPCR is a common method for detecting and quantifying gene expression, but the accuracy and reliability of the results are highly dependent on appropriate data normalization [24]. Recently, RNA-Seq has emerged as a powerful high-throughput technology used for transcriptome analysis of various organisms and treatments [24]. It can also be used to obtain an informative, holistic, and unbiased picture clarifying the hepatocyte differentiation process via liver cell models [13]. Revolutionized genomic and transcriptomic techniques show high potential in liver-related research because they are affordable, fast, and precise [25], and RNA-Seq analysis has been used previously to investigate hepatic lineage development [26]. Even though several liver cell models have been analyzed via RNA-Seq to compare liver cells in vitro and liver tissue in vivo [13], there are few studies examining small hepatocyte differentiation in vitro that use RNA-Seq analysis.

Hybrid periportal hepatocytes, which are a differentiation stage between oval cells and mature hepatocytes, show extensive proliferation abilities and are highly efficient in repairing livers that are deficient in healthy hepatocytes [27]. Oval cells are also known to differentiate into mature hepatocytes via small hepatocytes [23,28]. Although oval cells are mainly found in hepatotoxin-treated livers, small hepatocytes can be isolated from healthy livers [23]. Furthermore, oval cells are bipotential stem cells that can differentiate into a wide range of cells, including hepatocytes, bile epithelial cells, pancreatic cells, and intestinal epithelial cells [29]. We thus reasoned that small hepatocytes could be used to clarify the liver differentiation process in a specific, simple, and appropriate manner, and supported this idea by performing RNA-Seq analysis and comparing the fold change expression results with the qPCR analysis. The comparison of RNA-Seq and qPCR results using the same samples showed that the two methods yielded the same expression patterns.

### 3.1. Comparison between Mature Hepatocyte Cultures and Small Hepatocyte Cultures

Mature hepatocytes are still considered the gold standard for in vitro drug screening and toxicity studies because they can provide the complete complement of hepatic drug-metabolizing enzymes and transporters [15]. Nevertheless, mature hepatocytes typically undergo progressive dedifferentiation in three days when cultured as monolayers in vitro, which is reflected by their drug transporters and the dramatic loss of phenotypic characteristics [15]; we observed similar results (Figure 1). Mature hepatocytes obtained by liver resection gradually lost their original characteristics due to an epithelial–mesenchymal transition from the originally functional mesenchymal type (Figure 1A,B) to the epithelial type (Figure 1C,D), which has also been reported previously [30,31]. Because attempts to proliferate mature hepatocytes in vitro have been less successful despite their prolific growth abilities in vivo, it has taken a long time to be able to grow them steadily by optimizing mature hepatocyte culture conditions [15]. Mature hepatocytes can turn into fibroblast-like cells after five days of in vitro culturing, and gradually but eventually lose their functionality. In contrast, small hepatocytes have been identified as proliferating cells with hepatic characteristics that show a remarkable ability to clonally proliferate into colonies and differentiate in vitro (Figure 1E–H); they can also stay alive for at least 35 days of culturing (Figure 2). Therefore, small hepatocytes are a potential cell source for in vitro hepatocyte cultures due to their ability to mimic in vitro liver tissue.

The correlation plot of transcriptome pairs (Figure 3) shows that culturing cells in vitro has a considerable effect on gene expression. The expression of the transcriptome is expected to be specific; in other words, the number of expressed transcripts should be gradually getting lower in cultured hepatocytes. The gene expression of Rat B cells (cultured from 7 to 21 days) had a same tendency, indicating that the results were similar to what was expected, whereas Rat A cells did not exhibit the same trend towards mature hepatocyte differentiation. For this reason, the repeatability of the experiment was called into question because the sample from Rat A could be a failed culture. The results obtained by qPCR were very similar to those obtained by RNA-Seq for both Rat A and Rat B, confirming that RNA-Seq analysis can be used to complement qPCR results when investigating the hepatocyte differentiation process, particularly because RNA-Seq can identify genes that then can be examined using qPCR [32].

The expression levels of albumin and TO increased on day 15 of culturing, whereas the expression of FO decreased on day 15 in Rat B cells (Figure 4). Therefore, increased TO and reduced FO expression, which has also been observed in a previous study [1], might stimulate small hepatocyte differentiation into mature hepatocytes. The increased expression of albumin in Rat B cells demonstrated that small hepatocytes exhibited more hepatic functions after long-term culture. However, the differences between Rat A and Rat B gene expression results may have occurred because the culture conditions were unsuitable for cell proliferation, which resulted in the failure of small hepatocytes to differentiate into mature hepatocytes in Rat A cells. This could be because the differentiation protocols do not always result in mature hepatocytes that can perform hepatic functions such as albumin and urea secretion and drug metabolization [33].

Mature hepatocytes from both Rat A and Rat B were also unable to maintain a steady growth rate for extended periods of time, and even lost their function after 14 days of culturing. This might have been because as soon as the mature hepatocytes became stable, sinusoid phenotype-maintaining factor secretion resumed, and the secretion and synthesis of laminin terminated. This interplay between the extracellular matrix (ECM) and hepatocytes may have contributed to disturbed hepatic functioning [34]. The ECM has been shown to enhance hepatocyte attachment, but this usually occurs concomitantly with hepatocyte spreading and a subsequent loss of hepatic function [35,36]. However, the RNA-Seq analysis was consistent with qPCR results for both Rat A and Rat B, indicating that RNA-Seq is helpful for acquiring the expression levels of all transcripts in a single run, and can serve as an efficient method to validate success rates for biological experiments.

### 3.2. Comparison of RNA-Seq-Specific Transcript Expression Changes with qPCR and RNA-Seq Analysis

We employed RNA-Seq to characterize the comprehensive transcriptional profiles of small hepatocytes selected from several stages of differentiation between days 7, 15, and 21. Figure 4 compares the differences between qPCR (∆∆Cq Expression) and RNA-Seq (transcript per million [TPM]) analyses. qPCR results are shown with a ratio plot based on the gene expression profile on day seven, and TPM was used as a benchmark for RNA-Seq, also based on results from day seven. We found that the ratio of the transcript expression levels of each target gene was consistent across different time points, which proved that the results of the RNA-Seq analysis could be used for target-specific selection. These results are consistent with results from traditional biological experimental processes, verifying the state of the differentiating cells.

### 3.3. Comparison of DAVID Analysis with Cell Morphology and qPCR Analysis

Small hepatocytes tended to form dense spheroid colonies, but mature hepatocytes easily grew in monolayers. Results from an analysis using DAVID [37] indicated a similar trend: genes related to cell–cell adherence and cell–cell interactions were more highly expressed in small hepatocytes, whose spheroid colonies exhibited noticeable cell interactions (Figure 2B). Moreover, the qPCR results showed more metabolism-related functions associated with albumin and TO in the small hepatocyte culture with time (Figure 4). The results in Appendix A also show that mature hepatocytes had higher expression levels of genes related to metabolic processes. Liver progenitor cells can express surface markers that are characteristic of immature hematopoietic cells, which may contribute to the development of resident liver immune cell populations through local hepatic immune cell differentiation [38]. Furthermore, liver progenitor cells express the epithelial cell adhesion molecule (EpCAM), which is involved in cell–cell adhesion and signaling transduction and is absent from mature hepatocytes [39,40]. The results from these previous studies were consistent with our findings (Appendix A).

### 3.4. Gene Enrichment Test

The transcript expression levels (TPM) of the top 20 terms from the gene enrichment analysis are shown in Appendix A. These tables demonstrate the success of Rat A and Rat B cultures from day 7 to 21. Because small hepatocyte gene expression was initially upregulated compared to mature hepatocytes (the list of Appendix A and longer than that of Appendix A), small hepatocyte gene expression levels were expected to decrease from day 7 to 21. However, by observing the numbers of expressed mRNA (NM) in Table 3, Rat A showed an upward trend, which was consistent with the previously observed culture failure (Figure 4). Furthermore, we used all listed upregulated DEGs that were consistent with the results of the top 10 terms from the gene enrichment analysis. These results suggested that the difference in transcript expression levels obtained by differential expression analysis can be used to determine the success of the mature hepatocyte cultures. The 836 downregulated transcripts can potentially be used to distinguish stem cells (liver progenitor cells) from liver cells (mature hepatocytes) (Figure 5). Subsequent biological experimentation is needed to verify the availability of these transcripts. In Figure 6, the average expression does not change much in both Rat A and Rat B, possibly due to the presence of false positives from low-expression transcripts.

Our study compared qPCR and RNA-Seq analysis for small hepatocyte cultures and found similar expression patterns. RNA-Seq offers several advantages, such as low RNA requirement, high reproducibility, ability to detect mutations and alternative transcripts, and the fact that a single experiment can provide information about all genes [32]. Although further investigation is needed to assess the potential for replacing qPCR with RNA-Seq, these complementary techniques are useful for understanding dynamic liver development processes. Additionally, our results highlight the importance of utilizing high-throughput transcriptomic data for selecting appropriate reference genes for qPCR analysis [24].

In summary, our results provide novel insights into the differentiation and maturation of small hepatocytes into mature hepatocytes using RNA-Seq analysis. We identified small hepatocytes from developing rat livers through marker-free transcriptomic profiling, which may aid in the identification of biomarkers for isolated liver progenitor cells. However, translating in vitro data into reliable predictions applicable to human body responses remains largely undefined [41]. Developing predictive computational models for liver developmental processes and hepatic metabolism is crucial, alongside in vitro efforts [41]. Our future work will investigate the use of a microfluidic system with small hepatocytes for real-time genomic, transcriptomic, and epigenomic studies. In addition, we plan to further explore the expression of genes known to be influenced by culture conditions, such as drug metabolism enzymes, including cytochromes P450 and oxidation/reduction pathways, in future experiments. Moreover, we aim to identify additional factors involved in the differentiation of stem cells into hepatocytes by collecting more experimental data using RNA-Seq. The integration of genetic and in vitro studies could accelerate liver-related research in the near future.

### 3.5. Limitations of the Study

To improve the reliability and accuracy of our study’s summary statistics, it is necessary to evaluate the consistency of biological replicates with more than two replicates. Furthermore, to better understand the differentiation trajectories of liver progenitor cells, single-cell RNA sequencing should be used to explore the liver’s complex systems beyond different cell types and to build on previous research that revealed the heterogeneity of primary hepatocytes and other liver nonparenchymal cells.

## 4. Materials and Methods

### 4.1. Chemicals and Equipment

Chloroform (288306), L-ascorbic acid (A8960-5G), L-proline (P0380-100G), dexamethasone (D4902-100MG), picrylsulfonic acid solution (P2297-10ML), hydrochloric acid (30721-1L), ethanol (32221-2.5L), and acetone (32201-2.5L) were purchased from MilliporeSigma (Munich, Germany). 2-propanol (29113-95), nicotinamide (24317-72), albumin bovine serum (08587-42), sodium chloride (31320-05), potassium chloride (28514-75), sodium phosphate monobasic dihydrate (317-18), sodium phosphate dibasic dodecahydrate (31723-35), and phenol red (26807-21) were purchased from Nacalai Tesque (Kyoto, Japan). Trypan blue 0.4% (207-17081) and collagenase (034-22363) were purchased from Fujifilm Wako Pure Chemicals (Osaka, Japan). TRIzol reagent (15596018), FBS (26140-079), penicillin-streptomycin (15140-122), a ReadyProbes cell viability imaging kit (Blue/Green) protocol (R37609), cytokeratin 18 antibody (MA1-06326), goat anti-mouse IgG antibody (A-11001), UltraPure DNase/Rnase-free distilled water (10977-015)*,* ITS (51500-056), goat anti-rabbit IgG antibody (A11012), prolong gold antifade mountant with DAPI (P36931), and HEPES (172571000) were purchased from Thermo Fisher Scientific (Waltham, MA, USA). A PrimeScript 1st strand cDNA synthesis kit (6110A) was purchased from TaKaRa (Shiga, Japan). Rotor-Gene SYBR Green PCR master mix 2× was purchased from QIAGEN (Hilden, Germany). Cellmatrix type I-A 3 mg/mL (Collagen; 160222) was purchased from Nitta Gelatin (Osaka, Japan). A QuantiChrom urea assay kit (DIUR-100) was purchased from BioAssay Systems (Hayward, CA, USA). A rat albumin ELISA quantitation kit (E110-125) was purchased from Bethyl Laboratories (Montgomery, TX, USA). DMEM/F12 (CC113-0500), WE medium (CC901-0500), and DMEM medium (CC103-0500) were purchased from Simply GeneDireX (Taoyuan, Taiwan). Epidermal growth factor (EGF; 354001) was purchased from Corning (Corning, NY, USA). Primers for qPCR were purchased from LGC Biosearch Technologies (Hoddesdon, UK).

A light microscope (Axio Vert. A1) was purchased from Carl Zeiss AG (Oberkochen, Germany). A fluorescence spectrometer (FP-8300) and a UV/Vis spectrophotometer (V-530) were purchased from JASCO (Tokyo, Japan). A microplate absorbance reader (Sunrise) was purchased from Tecan (Männedorf, Switzerland). A TurboCycler lite PCR thermal cycler (TCLT-9610) was purchased from Blue-Ray Biotech (Taipei, Taiwan). qPCR (Rotor-Gene Q) was purchased from QIAGEN (Hilden, Germany). The experimental protocol was reviewed and approved by the Ethics Committee on Animal Experiments of National Taiwan University (IACUC Approval No: NTU105-EL-00058).

### 4.2. Hepatocyte Isolation and Seeding

Mature hepatocytes and small hepatocytes were obtained from 6–8-week-old male Sprague Dawley rats (*Rattus norvegicus*) using a two-step collagenase perfusion method with some modifications, as described previously [42,43]. The isolated mature and small hepatocytes were seeded at a density of 2 × 10^5^ and 3 × 10^5^ cells/mL on 6 mm collagen-coated dishes, respectively. The mature hepatocytes were maintained in WE medium with 5% FBS. The medium was changed 4 h and 24 h after inoculation and every 48 h thereafter. The small hepatocytes were maintained in DEME/F-12-based medium (500 mL DEME/F-12 medium containing 15 mg L-proline, 4 mL penicillin-streptomycin, 1.67 mL 30% BSA solution, 5.5 mL of 1 M nicotinamide solution, 5 mL of 100 mM L-ascorbic acid, 5 mL of ITS solution, 0.5 mL of 10 μg/mL EGF, 0.5 mL of 10^−4^ M dexamethasone solution, and 0.5 mL 50 mg/mL gentamicin solution, sterile-filtered into the solution with a 0.22 μm filter). The medium was changed 4 h and 24 h after inoculation and every 3 days thereafter. Both mature hepatocyte and small hepatocyte cultures were maintained at 37 °C under 5% CO_2_ in a humidified incubator. Hepatocyte morphology was investigated using microscopy. Small hepatocyte cell viability was investigated using a live/dead fluorescence assay. The dishes were rinsed with PBS and incubated with a ReadyProbes Cell Viability Imaging Kit (Blue/Green) for 15 min at 37 °C after 12 and 35 days of culturing.

### 4.3. Total RNA Extraction and Reverse Transcription

We extracted total RNA from the cultured hepatocytes using TRIzol reagent, as previously described [44]. The RNA quality was verified by its OD260:OD280 absorption ratio. The total RNA (500 ng) isolated from hepatocytes was reverse-transcribed into single-strand cDNA using PrimeScript Rtase and 50 pM random hexamer primers. PCR was performed using primers for hepatic function markers (albumin, FO, and TO), and actin was used as a housekeeping gene. These primers are listed in Appendix A.

### 4.4. Real-Time PCR

For the Rotor-Gene SYBR Green PCR reaction, 8 μL RNAse-free water (PCR grade), 0.5 μL forward primer (20 μM), 0.5 μL reverse primer (20 μM), and 10 μL Rotor-Gene SYBR Green Master were mixed and prepared. cDNA (1 μL) was amplified using a standardized PCR protocol with the Rotor-Gene SYBR Green PCR kit. The melting curve analysis program from Rotor-Gene Q was used to identify specific PCR products. Quantitative gene expression data were normalized to the expression levels of actin. Gene expression values of each sample om days 7, 15, and 21 were first normalized to the respective expression values of actin, and further normalized to day 7 values for comparison. Relative gene expression was then calculated as fold change. 

### 4.5. RNA-Seq Sample Preparation

The samples used in the RNA-Seq analysis are displayed in Table 1. In total, four rats were sacrificed for sample preparation. A sample of mature hepatocytes (M1) was acquired from Rat a, and Rat b served as a biological replicate (M2) for the mature hepatocyte sample. From Rat b, two samples of small hepatocytes (S0-1 and S0-2) were harvested. We treated these two samples as technical replicates to validate the quality of the RNA-Seq data. Two samples of small hepatocytes cultured in vitro on day 7 were from Rat A (S7-1 and S7-2); again, S7-1 and S7-2 are technical replicates. These six samples (M1, M2, S0-1, S0-2, S7-1, and S7-2) were sent for sequencing as a batch. Another sequencing batch contained six more samples: S7-A, S7-B, S14-A, S14-B, S21-A, and S21-B, where S7-A, S14-A, and S21-A were acquired from Rat A and S7-B, S14-B, and S21-B were from Rat B. The sample S7-A was designed to be compared with S7-1 and S7-2 to monitor the batch effect. Among the second batch, S14-A and S14-B were two biological samples of small hepatocytes cultured in vitro on day 14, and S21-A and S21-B were two biological samples of small hepatocytes cultured in vitro on day 21. For all twelve samples, after total RNA extraction and Dnase I treatment, magnetic beads with oligo dT were used to isolate mRNA (for eukaryotes). After qPCR, the sequencing step used Illumina HiSeq 4000 to generate raw reads of samples.

### 4.6. Overview of RNA-Seq Analysis

The adopted analysis procedure is shown in Figure 7 The analysis was divided into six steps: (1) check the quality of raw data, (2) download reference transcriptome, (3) read mapping, (4) expression quantification, (5) differential analysis, and (6) gene enrichment and annotation. Detailed explanations and related tools are described in the following sections.

### 4.7. Checking the Quality of Raw Data

The RNA-Seq data used in this analysis were divided into five groups according to tissue and processing conditions: (1) mature hepatocytes retrieved from the rats without culturing (M1 and M2), (2) small hepatocytes cultured in vitro on day 0 (S0-1 and S0-2), (3) small hepatocytes cultured in vitro on day 7 (S7-1, S7-2, S7-A, and S7-B), (4) small hepatocytes cultured in vitro on day 14 (S14-A and S14-B), and (5) small hepatocytes cultured in vitro on day 21 (S21-A and S21-B). The sequencing data were paired-end, and the reads were strand-specific with a reverse–forward direction. The sequence length of the reads was 150 bp, and the number of read pairs was between 24M and 25M in each sample. After initial quality control using FastQC, the per-unit sequencing quality analysis charts showed that almost all the reads fell within the confidence interval (Phred quality score between 28 and 40). To retain more information, the 12 sets of sequencing data were subsequently analyzed without filtering or trimming.

### 4.8. Downloading the Reference Transcriptome

For the next-generation sequencing process (Figure 7), to determine the intensity of gene expression, we needed a set of transcript sequences that could be mapped to the reads. The rat transcripts in RefSeq [45] provided by the NCBI GeneBank database were used as a reference set (https://www.ncbi.nlm.nih.gov/assembly/GCF_000001895.5, accessed on 9 January 2019).

### 4.9. Read Mapping

Mapping can determine a preliminary amount of expression after the reads are mapped to the transcript sequences. In this step, the choice of mapping tool algorithm affects the subsequent quantification. As the default tool for RNA-Seq by RSEM [46], which is a widely-used method for expression quantification, Bowtie2 [47] was chosen to perform sequence mapping in this study.

### 4.10. Expression Quantification

Since genes may have alternative spliced transcribed forms, it is easy to estimate gene expression levels with errors by simply counting the number of reads of a particular gene. Here, we used RSEM to estimate the expression intensity of different gene transcripts. RSEM establishes a mathematical model of the maximum likelihood estimation with the expectation–maximization algorithm for expression estimation. In this study, RSEM was used after read mapping using Bowtie2.

### 4.11. Differential Analysis

DESeq2 [19] utilizes a negative binomial distribution to model gene counts and identifies differentially expressed genes (DEGs) based on their expression levels across distinct conditions. These genetic changes with differences in expression are likely related to the genes that drive small hepatocytes to become mature hepatocytes. Information relating to DEGs can be used to understand the transition from small hepatocytes to mature hepatocytes.

### 4.12. Gene Enrichment and Annotation

In this experiment, the transcript of a rat (*Rattus norvegicus*) was subjected to annotation analysis using the open annotation search tool DAVID (The Database for Annotation, Visualization, and Integrated Discovery) [37]. After comparing the target gene lists with the rat background set, the enriched input genes’ GO (Gene Ontology) [20] terms were identified and reported. Functional categories of BP_ALL (biological processes), CC_ALL (cellular components), and MF_ALL (molecular functions) are included for GO term annotation.

### 4.13. Statistical Analysis

To identify DEGs, we applied Bonferroni [48] with an adjusted *p*-value threshold of <1 × 10^−5^ in DESeq2 for multiple hypothesis testing. To determine the GO terms associated with the top enriched genes, Fisher’s exact test [49] was performed separately for the upregulated and downregulated DEGs. The top 20 GO terms ranked by their *p*-values are listed in Appendix A.

## 5. Conclusions

Our study demonstrated the concordance between RNA-Seq and qPCR gene expression measurements while highlighting the comprehensive view provided by RNA-Seq. The small hepatocyte-based liver cell model closely resembled the in vivo hepatocyte environment, enabling predictive and quantitative identification of differentiated hepatocytes. Our findings shed light on liver model differentiation and maturation, and highlight the potential of our system for liver disease monitoring and diagnosis in medical applications.

## Figures and Tables

**Figure 1 ijms-24-07534-f001:**
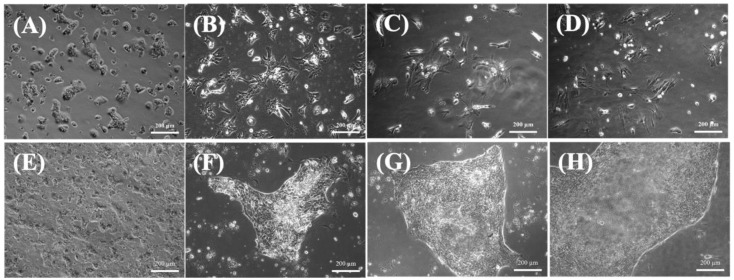
Phase-contrast photographs of hepatocytes on a collagen-coated dish. (**A**) Mature hepatocytes on day 1. (**B**) Mature hepatocytes on day 3. (**C**) Mature hepatocytes on day 5. (**D**) Mature hepatocytes on day 7. (**E**) Small hepatocytes on day 3. (**F**) Small hepatocytes on day 9. (**G**) Small hepatocytes on day 15. (**H**) Small hepatocytes on day 21. Scale bar: 200 μm.

**Figure 2 ijms-24-07534-f002:**
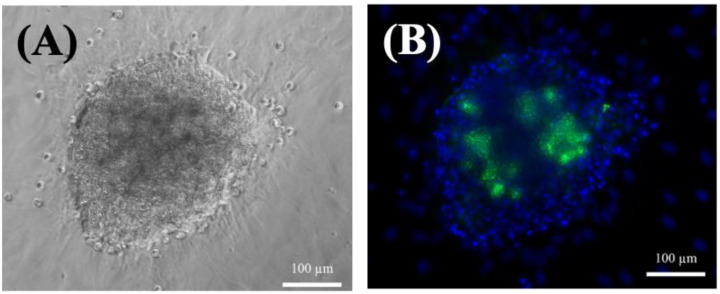
(**A**) Phase-contrast photographs of small hepatocytes on day 35. (**B**) Fluorescence images of small hepatocytes on day 35. The cells were cultured on a collagen-coated dish. Blue: DAPI; Green: NucGreen^®^ Dead reagent. Scale bar: 100 μm.

**Figure 3 ijms-24-07534-f003:**
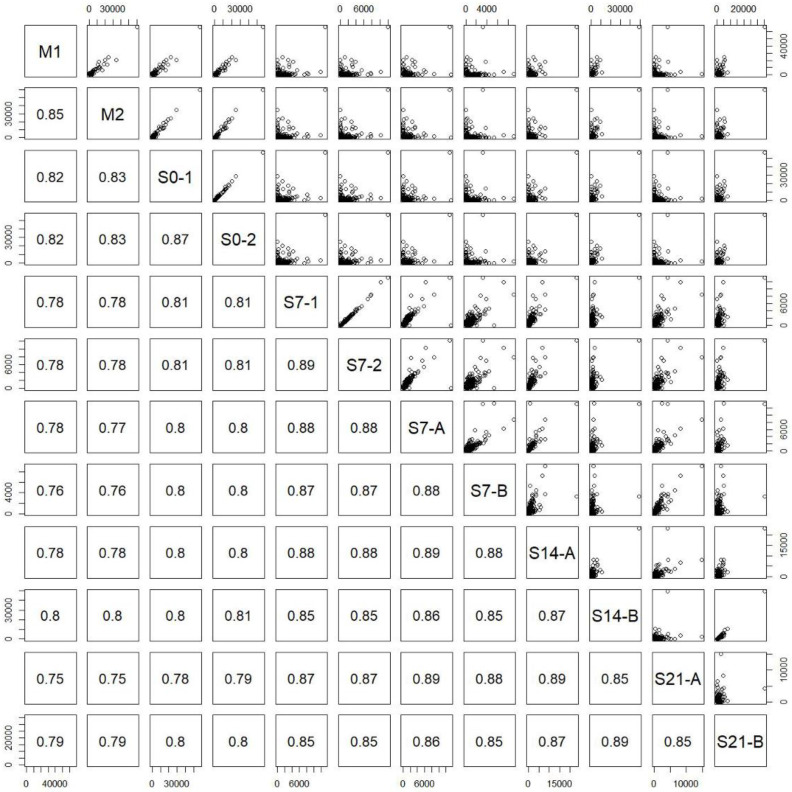
Correlation plots for each pair of samples. M1 and M2 were biological replicates; S0-1 and S0-2 were technical replicates; S7-1 and S7-2 were technical replicates; S7-A and S7-B, S14-A and S14-B, and S21-A and S21-B were biological replicates, respectively. The first six samples (M1, M2, S0-1, S0-2, S7-1, and S7-2) were sequenced in the 1st batch, and the rest were sequenced in the 2nd batch. S7-1, S7-2 and S7-A were technical replicates, but the sample S7-A was designed to be compared with S7-1 and S7-2 to monitor the batch effect.

**Figure 4 ijms-24-07534-f004:**
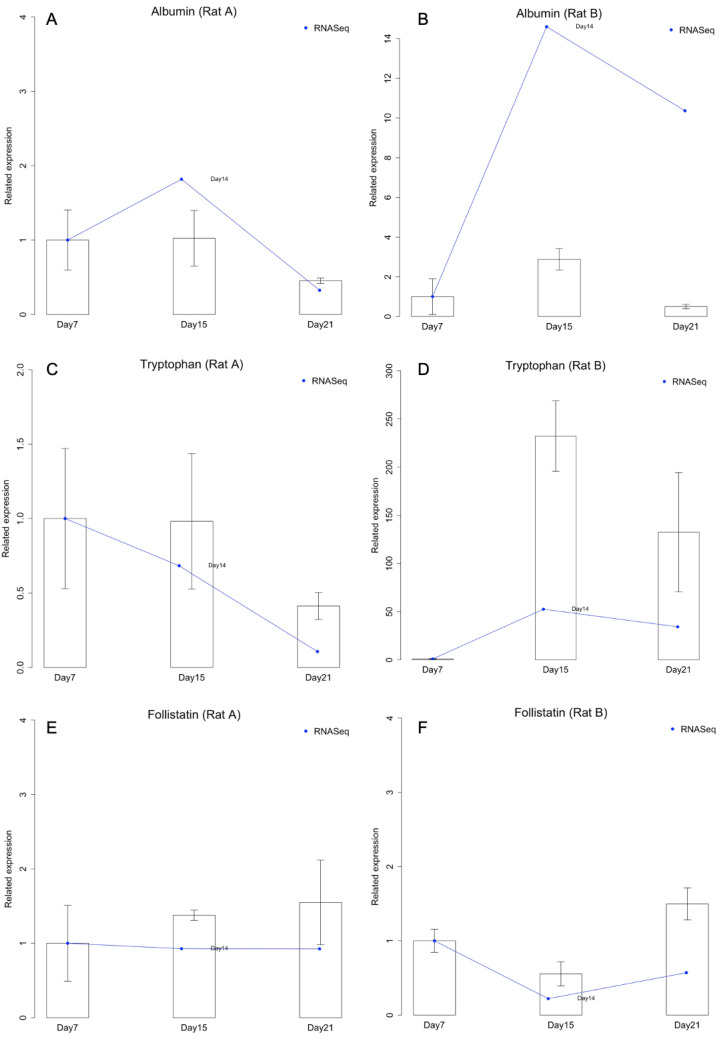
Comparison of the expression intensities quantified by RNA-Sequencing (RNA-Seq) and quantitative polymerase chain reaction (qPCR). (**A**) Albumin expression in Rat A. (**B**) Albumin expression in Rat B. (**C**) Tryptophan expression in Rat A. (**D**) Tryptophan expression in Rat B. (**E**) Follistatin expression in Rat A. (**F**) Follistatin expression in Rat B.

**Figure 5 ijms-24-07534-f005:**
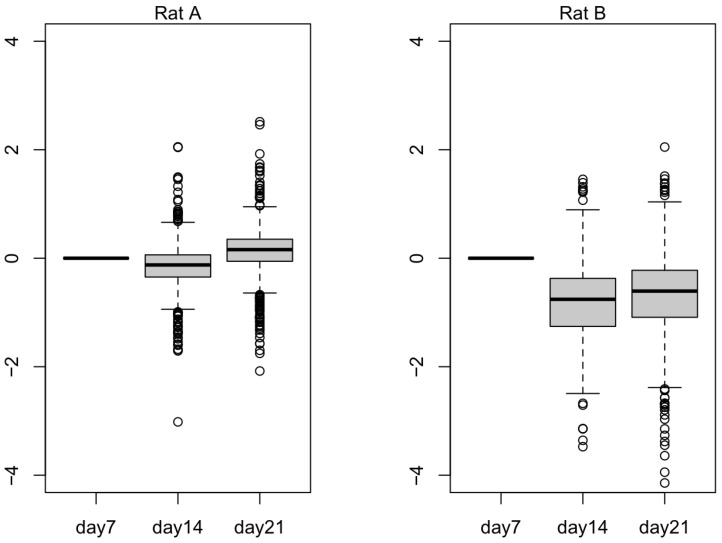
The log expression level of 836 downregulated differentially expressed genes (DEGs) from Rat A and Rat B. Outliers that fall below the first quartile minus 1.5 times the interquartile range or above the third quartile plus 1.5 times the interquartile range are represented as hollow circles in the plot.

**Figure 6 ijms-24-07534-f006:**
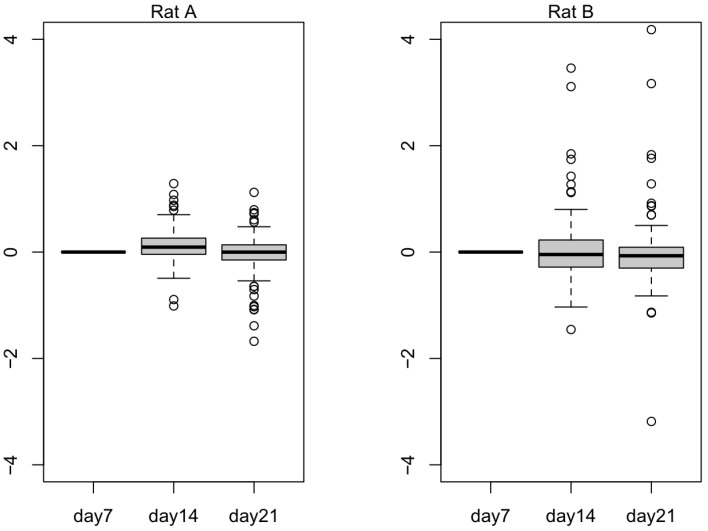
The log expression level of 137 upregulated differentially expressed genes (DEGs) from Rat A and Rat B. Outliers that fall below the first quartile minus 1.5 times the interquartile range or above the third quartile plus 1.5 times the interquartile range are represented as hollow circles in the plot.

**Figure 7 ijms-24-07534-f007:**
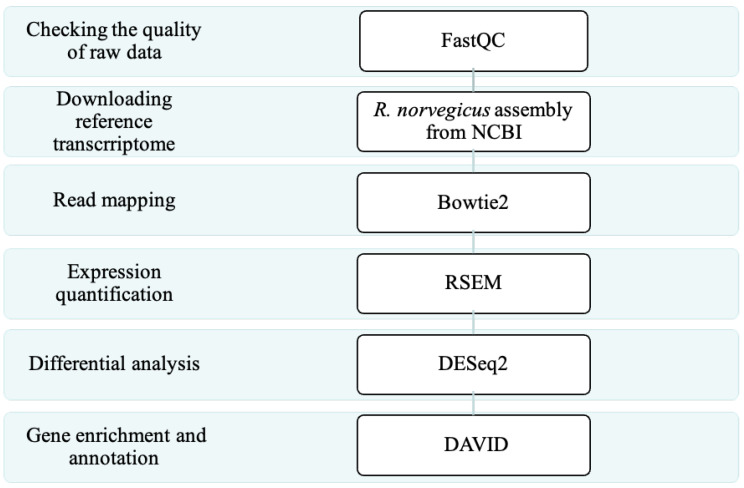
The next-generation sequencing (NGS) pipeline adopted in this study.

**Table 1 ijms-24-07534-t001:** The twelve RNA-seq samples produced in this study.

Sample ID	Rat a	Rat b	Rat A	Rat B
Mature_1 (M1)	●			
Mature_2 (M2)		●		
Small_day0_1 (S0-1)		●		
Small_day0_2 (S0-2)		●		
Small_day7_1 (S7-1)			●	
Small_day7_2 (S7-2)			●	
Small_day7_A (S7-A)			●	
Small_day7_B (S7-B)				●
Small_day14_A (S14-A)			●	
Small_day14_B (S14-B)				●
Small_day21_A (S21-A)			●	
Small_day21_B (S21-B)				●

**Table 2 ijms-24-07534-t002:** Mapping information of each sample.

Sample ID	Number of Reads with Unique Alignment	Total Number of Reads	Alignment Rate
M1	19,092,670	24,186,428	78.9%
M2	19,908,323	24,478,483	81.3%
S0_1	18,478,527	24,612,029	75.1%
S0_2	17,836,747	24,097,733	74.0%
S7_1	17,676,527	24,527,137	72.1%
S7_2	17,947,167	24,494,364	73.3%
S7_A	22,805,333	29,168,165	78.2%
S7_B	24,122,435	29,803,234	80.9%
S14_A	22,294,554	28,731,946	77.6%
S14_B	22,933,655	29,472,723	77.8%
S21_A	27,521,598	34,985,276	78.7%
S21_B	24,187,360	29,673,865	81.5%

**Table 3 ijms-24-07534-t003:** Quantification information of each sample.

Sample	# of Expressed Transcripts	# of Expressed XM	# of Expressed NM	# of Expressed XR	# of Expressed NR
M1	28,121	15,560	10,609	1866	86
M2	27,520	14,898	10,633	1903	86
S0_1	31,469	17,439	11,453	2484	93
S0_2	31,707	17,686	11,460	2468	93
S7_1	34,128	19,367	11,774	2881	106
S7_2	24,099	19,333	11,785	2876	105
S7_A	35,300	19,861	12,037	3280	122
S7_B	36,029	20,450	12,125	3338	116
S14_A	35,669	20,235	11,992	3327	115
S14_B	34,649	19,523	11,744	3268	114
S21_A	36,776	20,881	12,289	3484	122
S21_B	34,605	19,521	11,812	3157	115

Note: NM: mRNA; NR: ncRNA; XM: predicted mRNA model; XR: predicted ncRNA model: #: the number.

## Data Availability

The RNA-seq data generated during this study are available at Sequence Read Archive with accession numbers of SRA:SRR11671311, SRR11671312, SRR11671313, SRR11671314, SRR11671315, SRR11671316, SRR11671317, SRR11671318, SRR11671319, SRR11671320, SRR11671321, SRR11671322.

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
