# Peer review of "Monitoring Cultured Rat Hepatocytes Using RNA-Seq In Vitro"

_ijms, 2023, doi:10.3390/ijms24087534_

Round 1

Reviewer 1 Report

Dear authors,

After reviewing, I have several comments: more numeric details are necessary in the abstract; a statistical section is required for the Materials and Methods; the control should be more clearly detailed in the paper; section 4.5 - why only ”two biological replicates”.

Best regards!

Reviewer 2 Report

General comment

This manuscript is interesting and on the whole enjoyable to read. The introduction is clear and well written. It helps to understand the purpose of the study.

The presentation of the material and methods and the results needs to be improved as the Figures and Tables are not referenced in the text in ascending order. Some of the Figures and Tables are also disproportionately large in relation to their interest (Figure 7, Table 1, Figure 3, Table 2, Figure 5 and 6).

My main criticism/comment concerns the choice of what was presented in the comparison between the two analytical methods. Only albumin, tryptophan and follistatin were compared. It seems to me that it would have been interesting to also make this comparison on other genes whose expression is known to be affected by culture, for example drug processing enzymes, and in particular cytochromes P450 or oxidation/reduction processes, in connection with what is said in discussion L417-423.

Specific comments

Number the Figures and Table in ascending order. Figure 7 should be Figure 1, Table 3 should be Table 1...

Figure 4: add the mode of expression of the results (mean +/- SD?) and probably the SDs on the RNA Seq

L243: Explain abbreviation: DAVID: Data base for Annotation, Visualization, and Integrated Discovery

L524-571: a bit too long
